# *Escherichia coli* Strains Isolated from American Bison (*Bison bison*) Showed Uncommon Virulent Gene Patterns and Antimicrobial Multi-Resistance

**DOI:** 10.3390/microorganisms12071367

**Published:** 2024-07-03

**Authors:** Jonathan J. López-Islas, Daniel Martínez-Gómez, Wendy E. Ortiz-López, Tania Reyes-Cruz, Andrés M. López-Pérez, Carlos Eslava, Estela T. Méndez-Olvera

**Affiliations:** 1Doctorado en Ciencias Agropecuarias, Universidad Autónoma Metropolitana, Calzada del Hueso 1100, Villa Quietud, Coyoacán, Ciudad de México 04960, Mexico; jonathan.66mvz@gmail.com; 2Departamento de Producción Agrícola y Animal, Universidad Autónoma Metropolitana, Calzada del Hueso 1100, Villa Quietud, Coyoacán, Ciudad de México 04960, Mexico; dmartinez@correo.xoc.uam.mx (D.M.-G.); biol.wendy.ortiz@gmail.com (W.E.O.-L.); reyescruztania.biol@gmail.com (T.R.-C.); 3Red de Biología y Conservación de Vertebrados, Instituto de Ecología, A.C., Carretera Antigua a Coatepec 351, El Haya, Xalapa 91073, Mexico; andres.lopez@inecol.mx; 4Unidad Periférica Investigación Básica y Clínica de Enfermedades Infecciosas, Facultad de Medicina, UNAM—Hospital Infantil de México Federico Gómez, Cuidad de Mexico 06720, Mexico; eslava@unam.mx

**Keywords:** *Escherichia coli*, zoonoses, *Bison bison*, Shiga-Toxigenic *Escherichia coli*, drug resistance

## Abstract

**Simple Summary:**

Wildlife plays an important role in global health, especially considering that most emerging infectious diseases originate from these animals. The emergence of these diseases involves dynamic ecological interactions among wildlife, livestock, and human populations. During this process, non-pathogenic bacteria can acquire genes that encode virulence factors or antimicrobial resistance mechanisms through horizontal gene transfer. Since wild animals can serve as sentinels for new bacterial pathogens resulting from this process, analyzing microorganisms isolated from wildlife could help describe the molecular characteristics associated with virulence evolution. *Escherichia coli* is a Gram-negative commensal bacterium found in the intestinal tracts of almost all vertebrate species. However, several strains can cause diverse intestinal and extraintestinal diseases in animals, including humans. The results of this study show that American bison populations can harbor atypical virulence gene profiles (virotypes) of *E. coli*, indicating that natural environments can be crucial sites for studying virulence evolution.

**Abstract:**

*E. coli* is considered one of the most important zoonotic pathogens worldwide. Highly virulent and antimicrobial-resistant strains of *E. coli* have been reported in recent years, making it essential to understand their ecological origins. In this study, we analyzed the characteristics of *E. coli* strains present in the natural population of American bison (*Bison bison*) in Mexico. We sampled 123 individuals and determined the presence of *E. coli* using standard bacteriological methods. The isolated strains were characterized using molecular techniques based on PCR. To evaluate the diversity of *E. coli* strains in this population, we analyzed 108 suggestive colonies from each fecal sample. From a total of 13,284 suggestive colonies, we isolated 33 *E. coli* strains that contained at least one virulence gene. The virotypes of these strains were highly varied, including strains with atypical patterns or combinations compared to classical pathotypes, such as the presence of *esc*V, *eae*, *bfp*B, and *ial* genes in *E. coli* strain LMA-26-6-6, or *stx*2, *eae*, and *ial* genes in *E. coli* strain LMA-16-1-32. Genotype analysis of these strains revealed a previously undescribed phylogenetic group. Serotyping of all strains showed that serogroups O26 and O22 were the most abundant. Interestingly, strains belonging to these groups exhibited different patterns of virulence genes. Finally, the isolated *E. coli* strains demonstrated broad resistance to antimicrobials, including various beta-lactam antibiotics.

## 1. Introduction

*Escherichia coli* is a Gram-negative bacterium abundant in the gastrointestinal tract of many animal species, including humans. This bacterium is typically a harmless commensal member of the microbiota and lives with its host in a mutualistic relationship; however, some strains have developed the capability to cause diverse intestinal and extraintestinal diseases [1]. The pathogenicity of these *E. coli* strains depends on a wide range of virulence factors (VF) that confer various abilities to the pathogen, such as adhesion to epithelia, tissue invasion, iron acquisition, motility, and toxigenicity, among others. The acquisition of genes encoding these factors has led to different levels of virulence and has enhanced our understanding of the pathogenic mechanisms used by *E. coli* strains to infect various hosts. Pathogenic *E. coli* strains isolated from clinical cases in humans have been classified into (1) intestinal pathotypes: enterohaemorrhagic *E. coli* (EHEC), enteropathogenic *E. coli* (EPEC), enterotoxigenic *E. coli* (ETEC), enteroaggregative *E. coli* (EAEC), enteroinvasive *E. coli* (EIEC), and diffuse adherent *E. coli* (DAEC); and (2) extraintestinal pathotypes: uropathogenic *E. coli* (UPEC) and *E. coli* associated with meningoencephalitis (MNEC) [2]. Unfortunately, the origin, genetic structure, and virulence capabilities of *E. coli* strains recovered from free-range bison are unknown, and many questions about the biology of pathogenic variants remain unresolved. There are many reports about the presence of *E. coli* O157 in carcasses [3] and retail meat [4] but not in natural habitats. In retail beef, *E. coli* serotypes O121, O145, and O157 were identified, suggesting a possible risk of foodborne disease. Additionally, resistance to antimicrobial drugs has been reported in *E. coli* strains recovered from carcasses and fecal samples of Bisons [5,6].

In *E. coli*, conventionally, the phylogroup and serogroup have been associated with a specific virulence gene profile (virotype); for example, typical EPEC strains are distributed in two major phylogroups, B1 and B2, which further correspond to EPEC complexes 2 and 1, respectively. The *eae* gene type differentiates these two groups, the first group possessing the *eae* β type, while the second group has the *eae* α type. Atypical EPEC strains belong to phylogroup E and possess the *eae* γ gene. EHEC strains are distributed between groups A and B1, but the highest concentration of these pathovars is found in group E strains of serotype O157 [7]. This association is not clear in wildlife since virulence-associated genes seem to be independent of phylogroup and serotype [8]. An analysis of *E. coli* strains recovered from animals showed the presence of virulence genes in phylogroups A, B1, B2, C, D, E, and F, with a higher proportion in phylogroups A and B1, considered commensals [9]. Likewise, antimicrobial resistance in *E. coli* seems to change with host and habitat switching and does not depend on phylogroup and serogroup [10].

Although some characteristics of *E. coli* virotypes isolated from animals have been described using the ECOR collection (a set of reference strains of *E. coli*); unfortunately, this collection includes strains isolated principally from domestic and zoo animals, where possible cross-contamination with human strains could occur; also, this collection does not include *E. coli* strains recovered from Bison [7]. Isolation and characterization of *E. coli* strains recovered from wildlife are essential to understanding the diversity of this genus. A few studies have investigated pathogenic *E. coli* in wild bison despite evidence that they are asymptomatic carriers. A study conducted on farms and ranches in the Midwestern United States showed that the mean prevalence of *E. coli* O157 in bison was 47.4% [11]. Likewise, a study with wild ruminants showed that these animals represent significant reservoirs of *E. coli* containing the virulence-associated genes *stx*1 or *stx*2 (STEC). The frequency of STECs observed in wild ungulates was 19.4% (22.3% in elk and 15.3% in deer) [12]. Additionally, *E. coli* strains isolated from the feces of diarrheic and non-diarrheic water buffalo (*Bubalus bubalis*) showed diverse virulence factors such as *stx*1, *stx*2, *eae*, and EAST-1. The *E. coli* pathovars identified in diarrheic feces were ETEC, STEC, EPEC, and EHEC [13,14]. In this study, *E. coli* strains isolated from a population of American bison held in an ecological reserve in northwest México were analyzed to describe the antigenic, genetic, virulence, and resistance characteristics of strains present in free-ranging bison.

## 2. Materials and Methods

### 2.1. Study Area

This study was conducted at the ecological reserve “Rancho el Uno,” an area dedicated to the conservation of American bison (*Bison bison*). The ecological reserve is located within the Janos Biosphere Reserve (RBJ) in Chihuahua, México, in the northwest of Chihuahua. It is situated between meridians 108°56′49″ and 108°56′22″ West and parallels 31°11′7″ and 30°11′27″ North. The reserve covers an area of 5264 km^2^, with an altitude ranging from 1200 to 2700 m above sea level.

### 2.2. Experimental Design and Animal Management

One hundred twenty-three fecal samples from bison were collected during the Ecological Ranch’s annual medical supervision. This supervision involves transferring animals to a defined area, followed by their introduction into a squeeze chute for clinical evaluation. For fecal analysis, samples were taken directly from the rectum of the animals while they were inside the squeeze chute. Only animals older than one year were included in the study. The samples were placed in plastic containers and kept on ice until analysis in the laboratory.

### 2.3. Isolation and Identification of E. coli from Feces

Two grams of each of the 123 fecal samples collected were inoculated into 9 mL of buffered peptone water and incubated for 24 h at 37 °C. Then, 0.1 mL of the suspension was inoculated onto eosin methylene blue (EMB) agar (Oxoid, Lenexa, KS, USA) and incubated for 24 h at 37 °C. One hundred and eight green metallic sheen-like colonies from each sample were picked and identified using standard microbiology laboratory techniques. Isolates that tested positive for the methyl red test (Sigma-Aldrich, St. Louis, MO, USA), negative for the Voges-Proskauer test (MRVP broth, Oxoid, KS, USA), negative for the Simmons citrate test (citrate agar, Oxoid, KS, USA), negative for the hydrogen sulfide (H_2_S) test, positive for the indole test, positive for the motility test (SIM agar, Oxoid, KS, USA), negative for the urease production test (urea agar, Oxoid, KS, USA), and produced an acid-over-acid reaction in the Triple Sugar Iron (TSI) test (TSI agar, Oxoid, KS, USA) were subcultured on EMB agar (Oxoid, KS, USA) at 37 °C for 24 h and stored at 4 °C. This study used five *E. coli* strains provided by Dr. Carlos A. Eslava Campos from the Federico Gómez Children’s Hospital as controls. The five reference strains were as follows: EPEC O127, (E2348/69), EHEC O157, (EOL933), ETEC O78 (H10407), EAEC O42: NM, and EIEC O143: NM.

### 2.4. Virulence-Associated Genes Identification by PCR in E. coli Strains

All wildlife and control strains of *E. coli* were inoculated into 10 mL of Luria-Bertani broth and incubated for 24 h at 37 °C. The biomass was recovered via centrifugation at 1200× *g* for 15 min at 4 °C. For DNA extraction, the CTAB-NaCl method was used. The extracted DNA was then used for the amplification of seven virulence-associated genes by PCR (*eae*, *esc*V, *bfp*B, *stx*1, *stx*2, *st*, and *lt*). A commercial kit (PCR Master Mix 2X, Thermo Scientific, Waltham, MA, USA) was employed for the PCR assays, following the manufacturer’s protocol. Primer sequences and PCR conditions are provided in Appendix A. PCR products were analyzed via 1.8% agarose gel electrophoresis.

### 2.5. E. coli Serotyping and Genotyping

Strains identified as *E. coli* were serotyped using agglutination assays with 96-well microtiter plates and rabbit antisera against O1 to O187 somatic (O) antigens and 53 flagellar (H) antigens. The antisera were prepared in rabbits (SERUNAM, a registered trademark in México, with number 323158/2015) using a method previously reported [15]. Phylogenetic groups were determined via PCR multiplex using the primers described in Appendix A and under conditions detailed in a previously reported method. The designation of the phylogenetic groups (A, B1, B2, C, D, E, F, and *Escherichia* cryptic clade I) was established by the presence or absence of *chu*A, *yja*A, TspE4.C2, *trp*A and the *arp*A gene (Appendix A) [16].

### 2.6. Antimicrobial Resistance Evaluation in E. coli Strains

Susceptibility testing was performed on all *E. coli* strains. Conditions for the diffusion agar assays included commercial discs for antimicrobial testing (Investigación Diagnóstica, México) and Mueller Hinton agar plates (BBL). A standard 0.5 McFarland saline suspension of *E. coli* strains was seeded onto Mueller Hinton agar using a cotton swab. Then, a commercial disc was placed on the agar surface. The plates were then incubated at 37 °C overnight. Subsequent inhibition zones were measured with a caliper, and results were interpreted as recommended by the manufacturer. Susceptibility testing included a panel of twelve different antimicrobial drugs: cephalothin (30 μg), cefotaxime (30 μg), netilmicin (30 μg), ciprofloxacin (5 μg), norfloxacin (10 μg), chloramphenicol (30 μg), sulfamethoxazole and trimethoprim (25 μg), nitrofurantoin (300 μg), ampicillin (10 μg), amikacin (30 μg), carbenicillin (100 μg), and gentamicin (10 μg). A quality control strain, *E. coli* ATCC 25922, was used to validate the susceptibility testing procedure.

### 2.7. Statistical Analysis

Descriptive and comparative statistics were performed using R software (ver. 4.4.1.) *E. coli* frequencies were established according to individual, serotype, genotype, and each virulence gene.

## 3. Results

### 3.1. Isolation and Identification of E. coli from American Bison Feces

During 2015 and 2016, a total of 123 individuals were captured and examined, resulting in the collection of 123 fecal samples. From these, 13,284 suggestive colonies of *E. coli* were recovered, with 108 colonies isolated from each bison stool sample. All isolates were analyzed using standard procedures for bacterial identification and molecular techniques to identify virulence-associated genes (VAGs). *E. coli* strains carrying VAGs (*E. coli*-CVAGs) were found in only eight individuals (6.54%). From these animals, thirty-three distinct *E. coli*-CVAGs strains with at least one virulence gene were recovered from American bison (see Table 1). The frequency of virulence-associated genes showed that *stx*2 was the most common per individual (6 out of 8 bison) and overall (51.5%). The frequencies of the other virulence-associated genes were 33.3% for *eae*, 27.2% for *ial*, 18.1% for *stx*1 and *esc*V, and 9.09% for *bfp*B (see Table 1). These strains were serotyped and genotyped using standard methods.

### 3.2. E. coli Serotypes and Genotypes

Genotyping of the 33 *E. coli*-CVAGs strains indicated that all belonged to unrecognized genogroup. For these strains, PCR assays showed positive results for *arp*A, *yja*A, and TspE4.C2, which do not correspond to any pattern described by Clermont [16].

The serotyping of the *E. coli*-CVAGs strains showed that eight (24.2%) belonged to the O22 pathogenic serogroup, eight (24.2%) to the O26 pathogenic serogroup, three (9.0%) to the 112ab group, one (3.0%) to the O10 group, and six (18.1%) to both the O79 and O8 serogroups, respectively. Only one *E. coli*-CVAGs strain could not be typed.

The results for flagellar antigens showed that eight individuals (24.24%) were positive for the H14 antigen, seven (21.0%) for H8, four (12.12%) for H11, three (9.09%) for H2, and nine (27.27%) had negative results (see Table 1).

### 3.3. Antimicrobial Resistance in E. coli-CVAGs Strains

In agar diffusion assays, *E. coli*-CVAGs strains showed resistance to almost all antimicrobial drugs. The results indicated that all strains (100%) were resistant to chloramphenicol, netilmicin, nitrofurantoin, ampicillin, amikacin, carbenicillin, gentamicin, and cephalothin. Thirty-two strains (96%) were resistant to cefotaxime and ciprofloxacin, thirty-one (87.8%) to norfloxacin, and twenty-four (72.7%) to sulfamethoxazole and trimethoprim (see Table 2). Some *E. coli*-CVAGs strains resistant to antimicrobial drugs showed slightly different growth inhibition zones, suggesting varying growth capabilities in the presence of antimicrobial drugs.

## 4. Discussion

Although *E. coli* is an essential member of the normal intestinal microbiota, some strains represent significant pathogens of global concern. Animal sources of this bacterium are crucial in transmission, and increasing attention has been directed at wildlife reservoirs [14]. Isolation and characterization of pathogenic *E. coli* strains from animal hosts are essential to understand the diversity of this genus. In this research, 123 individuals were sampled to identify *E. coli*-CVAGs strains; the results showed that 6.54% carried this type of *E. coli*. A study conducted with farm-raised bison showed frequencies of EHEC O157 ranging from 17 to 83% [6], suggesting that in wild bison, the presence of *E. coli*-CVAGs was lower in comparison with farm-raised bison. However, the frequency of 6.54% obtained in this study is similar to that reported for EHEC O157 in domestic bovines. An investigation of cattle feces showed that out of 589 fecal samples, only 44 (7.5%) were positive for pathogenic *E. coli* [17]. Additionally, another study with fecal samples of sheep and goats showed a frequency of 4.7–8.6% [18]. The frequency rate observed in our study closely aligns with those reported in investigations involving domestic ruminants. This suggests that factors such as contaminated pasture might contribute to the high frequency of virulent *E. coli* observed in farm-raised American bison. Additionally, our data indicate that the occurrence of virulent *E. coli* in wild American bison is infrequent. However, this does not imply it is a minor public health concern.

The analysis of virulent genes revealed that *stx*2 was the most prevalent virulent gene, detected in 51.5% of individual and isolated strains. A previous study indicated that the *stx*2 gene was the most prevalent Shiga toxin gene in *E. coli* strains isolated from ungulates in Portugal [19]. Other studies in Europe have similarly reported a higher prevalence of *stx*2 in *E. coli* strains isolated from wild ungulates [20]. These findings suggest widespread dissemination of the *stx*2 gene in *E. coli* strains from wild ungulates. This is noteworthy, considering that all *E. coli* strains isolated in these studies belonged to different serogroups and genotypes and were recovered from natural environments. Interestingly, a study involving 150 *E. coli* strains isolated from bison found that *stx*1 and *eae* were the most frequent genes, with frequencies of 27.3% and 17.7%, respectively [9]. However, in that study, the animals were in contact with domestic cattle, which might explain this difference.

The genetic characterization of *E. coli*-CVAGs via PCR revealed different combinations for each virulent gene evaluated in this study. The virotypes identified in this study in wild Bison could be classified as STEC, EPEC, EHEC, and EIEC. An analysis of *E. coli* strains isolated from the feces of both diarrheic and non-diarrheic water buffalo calves also demonstrated diverse virulence factors, such as *stx*1, *stx*2, *eae*, and EAST-1. The *E. coli* pathovars identified in diarrheic feces included ETEC, STEC, EPEC, and EHEC [21]. Similarly, a previous study found that the pathotypes of *E. coli* strains isolated from bison calves were STEC, EPEC, ETEC, and EAEC [9]. These results confirm that wild ruminants are reservoirs of STEC strains.

A study suggests that the coexistence of specific virulence-associated genes (such as *eae* and *esc*V) offers an advantage for *E. coli* adaptation to the bovine intestinal environment. During these evolutionary processes, the presence of bacterivorous protozoa in the bovine intestine exerts selection pressure to maintain *stx* and other virulence-associated genes associated with anti-predation activities [22]. In our study, the most frequent virulent gene was s*tx*2 (51.5%), followed by *eae* (33.3%), *ial* (27.2%), *stx*1 (18.1%), *esc*V (18.1%), and *bfp*B (9.09%). Analysis of gene combinations showed that the *stx*2 gene often appeared alongside *eae*, stx1, and *ial* in *E. coli*-CVAGs.

In the present study, *E. coli*-CVAGs strains exhibited various serogroups associated with virulent strains, each containing at least one virulent gene. The most frequent serogroups were O26 and O22, both with a frequency of 24%. O26 strains are among the most reported non-O157 STEC strains associated with human infections worldwide [23]. Twelve serogroups have been described in EPEC: O26, O55, O86, O111, O114, O119, O125, O126, O127, O128, O142, and O158. These serogroups include both typical and atypical EPEC strains, as well as other diarrhoeagenic *E. coli* categories [1]. In this study, serogroup O26 was identified in *E. coli* isolates recovered from American bison. Interestingly, the serotypes O26:H- and H11 (isolated in this study) represent heterogeneous *stx*-producing serotypes that include different clones or genetic lineages. Some authors even classify them as EHEC or STEC. In this study, serogroup O26 displayed different virotypes and tested positive for *eae*, *esc*V, *ial*, and *bfp*B but not for *stx*. This suggests that the patterns of virulence genes in strains isolated from wildlife are different from those described in domestic animals or humans [23,24].

Similar to findings in other studies on American bison [5,6,25], *E. coli*-CVAG strains isolated from free-ranging bison exhibited multiple resistances to antimicrobials. A potential factor contributing to the increased multidrug resistance (MDR) levels could be the high or increasing degree of environmental anthropization. There is also considerable evidence that agricultural and waste management practices are leading to clinically relevant antimicrobial resistance dissemination through ecological pathways. Over the past several decades, wildlife has shown promise for evaluating the dissemination of antimicrobial resistance within the environment [26]. In this study, all strains were resistant to chloramphenicol, ampicillin, amikacin, carbenicillin, gentamicin, and cephalothin, suggesting possible environmental contamination. However, the distribution of other animals in the area, such as carnivores, could explain the presence of multi-resistant strains. Previous research has indicated that carnivorous animals are more frequently carriers of resistant strains than omnivorous animals [27]. Additionally, it has been reported that antibiotic-resistant strains are more commonly found in carnivores than in herbivorous animals [28]. Resistance to tetracycline and sulfamethoxazole is among the most common types of resistance observed in strains derived from wildlife [29].

Furthermore, a study demonstrated that cattle from the northeastern regions of Mexico are reservoirs of multidrug-resistant *E. coli*. In this study, the most common pathotype identified was EHEC (*stx*2- and *eae*-positive strains), followed by EPEC (*eae*- and *bfp*-positive strains), ETEC (*lt*- and *st*-positive strains), and EIEC (*ipa*H- and *vir*F-positive strains). Among the *E. coli* pathotype strains recovered in this study, there was significant resistance to erythromycin, trimethoprim/sulfamethoxazole, tetracycline, and β-lactams [30]. Additionally, another study of the antimicrobial resistance profile of *E. coli* strains isolated from bovine feces and carcass samples from Tamaulipas, Mexico, showed that 83.0% were resistant to ampicillin, 76.0% to cephalothin, and 69.0% to tetracyclines. Notably, the virulence-associated genes *eae* and *bfp* were not detected in any strain [13].

The detection of “unknown” phylogroup *E. coli* strains in this study was not an unexpected result. Despite the advantages of the quadruplex method used in this study, a small fraction of strains still needs to be assigned correctly to the appropriate phylogroup. The reasons for these failures could include the presence of extremely rare phylogroups or the occurrence of large-scale recombination events where the donor and recipient originated from different phylogroups [24]. The existence of unknown phylogroup *E. coli* strains uncovers the highly variable gene content driven by the frequent gain and loss of genes within wildlife strains. Studies of *E. coli* phylogroups in water buffalo have shown that many strains cannot be assigned to a specific phylogroup due to large-scale recombination between different phylogroups or the highly variable genome content driven by gene gain and loss [31].

The evolution of pathogen virulence in nature is a scientific question of great medical and biological importance. Using molecular techniques to detect genes encoding virulence factors aids in studying the different *E. coli* virotypes present in wild animals [32]. The identification of atypical combinations of virulence-associated genes in *E. coli* strains provides valuable insights into the evolution of new pathotypes, which may be more pathogenic and capable of causing diseases in animals. In this research, relevant characteristics of *E. coli* strains isolated from free-living bison are reported, providing important information on the epidemiology and attributes of this pathogenic microorganism in natural environments.

## 5. Conclusions

The analysis of *E. coli* strains isolated from wild animals is crucial for understanding the various characteristics of this pathogen. In this study, a community of American bison (*Bison bison*) hosted in an ecological reserve in northwest Mexico was analyzed to describe the genetic and virulent characteristics of *E. coli* strains present in free-range ruminants. Thirty-three *E. coli*-CVAGs strains were recovered from clinically healthy American bison. The results indicate that American bison carry atypical *E. coli* virotypes, suggesting the presence of potential pathogens in natural environments. However, more research is needed to establish the zoonotic potential of *E. coli*-CVAGs. Moreover, animal sources of *E. coli* are significant in the transmission of infections, and increasing attention has been directed towards wildlife reservoirs in recent years. However, the genetic analysis of *E. coli* strains isolated from wild American bison highlights the need to develop or expand classification systems, as *E. coli* variants isolated from wild animals exhibit different characteristics from those described in human strains.

## Figures and Tables

**Table 1 microorganisms-12-01367-t001:** Virotypes and serotypes of *E. coli*-CVAGs strains *.

Individual	*E. coli* Virotypes	Serotype	Strain Name
*B. bison*—3	*E. coli (escV*+*)*	O26 H11	LMA-26-2-1
	*E. coli (escV*+, *eae*+, *ial*+*)*	O26 H11	LMA-26-2-3
	*E. coli (escV*+, *ial*+*)*	O26 H11	LMA-26-6-22
	*E. coli (eae*+, *ial*+*)*	O26 H11	LMA-26-6-31
	*E. coli (ial*+*)*	O26 H-	LMA-26-2-5
	*E. coli (escV*+, *bfpB*+*)*	O26 H-	LMA-26-6-3
	*E. coli (escV*+, *eae*+, *bfpB*+, *ial*+*)*	O26 H-	LMA-26-6-6
	*E. coli (escV*+, *bfpB*+, *ial*+*)*	O26 H-	LMA-26-6-11
*B. bison*—17	*E. coli (stx1*+*)*	112ab H2	LMA-31-2-1
	*E. coli (stx1*+, *stx2*+*)*	112ab H2	LMA-31-2-32
	*E. coli (stx2*+*)*	112ab H2	LMA-31-2-34
	*E. coli (stx2*+*)*	O8 H14	LMA-31-2-4
	*E. coli (stx1*+*)*	O8 H14	LMA-31-2-8
	*E. coli (stx1*+, *eae*+*)*	O8 H14	LMA-31-2-16
	*E. coli (stx1*+, *stx2*+*)*	O8 H14	LMA-31-2-30
	*E. coli (eae*+*)*	O8 H14	LMA-31-3-24
*B. bison*—28	*E. coli (stx2*+*)*	O22 H8	LMA-3-6-1
	*E. coli (stx2*+, *ial*+*)*	O22 H8	LMA-3-6-5
*B. bison*—46	*E. coli (stx2*+*)*	O79 H14	LMA-49-1-2
	*E. coli (stx2*+, *eae*+*)*	O79 H14	LMA-49-1-7
	*E. coli (stx2*+*)*	O79 H-	LMA-49-1-9
	*E. coli (stx2*+, *eae*+*)*	O79 H-	LMA-49-1-28
	*E. coli (stx1*+*)*	O79 H-	LMA-49-1-33
*B. bison*—60	*E. coli (eae*+*)*	O8 H14	LMA-7-1-15
*B. bison*—63	*E. coli (stx2*+*)*	O? H21	LMA-15-3-3
*B. bison*—64	*E. coli (stx2*+, *ial*+*)*	O22 H8	LMA-16-1-28
	*E. coli (stx2*+, *eae*+, *ial*+*)*	O22 H8	LMA-16-1-32
	*E. coli (stx2*+*)*	O22 H8	LMA-16-3-18
	*E. coli (stx2*+*)*	O22 H-	LMA-16-3-33
*B. bison*—66	*E. coli (stx2*+*)*	O22 H8	LMA-14-3-1
	*E. coli (eae*+*)*	O22 H8	LMA-14-3–6
	*E. coli (eae*+*)*	O79 H7	LMA-14-3-9
	*E. coli (stx2*+*)*	O10 H-	LMA-14-3-12
	Total strains		33

* Only animals with *E. coli* strains tested positive for virulence-associated genes evaluated in this study were reported; commensal *E. coli* were found in all animals. “O?” indicates an unknown serogroup.

**Table 2 microorganisms-12-01367-t002:** Antimicrobial resistance of *E. coli*-CVAGs strains.

Strain Name	Antimicrobial Resistance											
	cfx	net	cpf	nof	cl	stx	Nf	am	ak	cb	ge	cf
LMA-26-2-1	R	R	R	R	R	R	R	R	R	R	R	R
LMA-26-2-3	R	R	R	R	R	R	R	R	R	R	R	R
LMA-26-2-5	R	R	R	R	R	R	R	R	R	R	R	R
LMA-26-6-3	R	R	R	R	R	R	R	R	R	R	R	R
LMA-26-6-6	R	R	R	R	R	R	R	R	R	R	R	R
LMA-26-6-11	R	R	R	R	R	R	R	R	R	R	R	R
LMA-26-6-22	R	R	R	R	R	R	R	R	R	R	R	R
LMA-26-6-31	R	R	R	R	R	R	R	R	R	R	R	R
LMA-31-2-1	I	R	R	I	R	S	R	R	R	R	R	R
LMA-31-2-4	R	R	R	R	R	R	R	R	R	R	R	R
LMA-31-2-8	R	R	R	R	R	I	R	R	R	R	R	R
LMA-31-2-16	R	R	R	I	R	I	R	R	R	R	R	R
LMA-31-2-30	R	R	R	R	R	R	R	R	R	R	R	R
LMA-31-2-32	R	R	I	R	R	I	R	R	R	R	R	R
LMA-31-2-34	R	R	R	R	R	R	R	R	R	R	R	R
LMA-31-3-24	R	R	R	R	R	R	R	R	R	R	R	R
LMA-3-6-1	R	R	R	R	R	R	R	R	R	R	R	R
LMA-3-6-5	R	R	R	R	R	R	R	R	R	R	R	R
LMA-49-1-2	R	R	R	R	R	R	R	R	R	R	R	R
LMA-49-1-7	R	R	R	R	R	R	R	R	R	R	R	R
LMA-49-1-9	R	R	R	R	R	I	R	R	R	R	R	R
LMA-49-1-28	R	R	R	R	R	R	R	R	R	R	R	R
LMA-49-1-33	R	R	R	R	R	I	R	R	R	R	R	R
LMA-7-1-15	R	R	R	I	R	I	R	R	R	R	R	R
LMA-15-3-3	R	R	R	R	R	R	R	R	R	R	R	R
LMA-16-1-28	R	R	R	R	R	R	R	R	R	R	R	R
LMA-16-1-32	R	R	R	R	R	R	R	R	R	R	R	R
LMA-16-3-18	R	R	R	R	R	R	R	R	R	R	R	R
LMA-16-3-33	R	R	R	R	R	R	R	R	R	R	R	R
LMA-14-3-1	R	R	R	R	R	R	R	R	R	R	R	R
LMA-14-3–6	R	R	R	R	R	R	R	R	R	R	R	R
LMA-14-3-9	R	R	R	R	R	R	R	R	R	R	R	R
LMA-14-3-12	R	R	R	I	R	I	R	R	R	R	R	R

Abbreviations: cefotaxime, cfx; netilmicin, net; ciprofloxacin, cpf; norfloxacin, nof; chloramphenicol, cl; sulfamethoxazole and trimethoprim, stx; nitrofurantoin, nf; ampicillin, am; amikacin, ak; carbenicillin, cb; gentamicin, ge; cephalothin, cf; resistant, R; intermediate resistance, I; susceptible, S.

## Data Availability

We are grateful to A. Vigueras, A. Rubio, and K. Moreno for helping during field sampling. We thank E. Ponce, R. Sierra (Janos Grassland Biological Station, IE-UNAM), and A. Esquer (Rancho El Uno TNC) for logistical support in the field.

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
