# Peer review of "Escherichia coli Strains Isolated from American Bison (Bison bison) Showed Uncommon Virulent Gene Patterns and Antimicrobial Multi-Resistance"

_microorganisms, 2024, doi:10.3390/microorganisms12071367_

Round 1

Reviewer 1 Report

Comments and Suggestions for Authors

The manuscript entitled "Escherichia coli strains isolated from American bison (Bison bison) showed uncommon virulent gene patterns and antimicrobial multi-resistance" was evaluated. 123 American bison (Bison bison) in Mexico had been involved in the investigation and 33 E. coli strains were isolated, then virotypes and genotypes of these strains were determined. A previously undescribed phylogenetic group has been revealed.In addition, E. coli strains demonstrated broad resistance to antimicrobials.This paper is in the scope of the journal.

Negative aspects

line 45:more keywords should be added.

Line183: the “Serotypes and” should be deleted as no interpretation in Table 2.

Line 211-215: : this paragraph should be combining with the previous paragraph.

Line 248-250: the reference should be included.

Line 259-272: the results should be comparing to the strains from humans in Mexico.

Line 273: “only” should be replaced with “one”.

Author Response

Reviewer 1

I appreciate your observations, and below I show you how they were addressed.

Line 45: more keywords should be added.

More keywords were already added.

Line 183: the “Serotypes and” should be deleted as no interpretation in Table 2.

“Serotypes and” was deleted in Table 2.

Line 211-215: this paragraph should be combining with the previous paragraph.

The paragraph was combined with the previous paragraph.

Line 248-250: the reference should be included.

The reference was already included.

Line 259-272: the results should be comparing to the strains from humans in Mexico.

A paragraph was added to describe multi-resistance in E. coli strains recovered from cattle farms contiguous to the study area. In addition, the text that talks about the modification of the environment was completed.

Line 273: “only” should be replaced with “one”.

This word was eliminated.

Reviewer 2 Report

Comments and Suggestions for Authors

This is a very interesting manuscript describing E. coli from bison. Some minor revision is needed prior publication.

As the paper is very short, please classy it a short article or communication.

Through-out the paper – instead of virulence genes – use virulence-associated genes (VAG), E. coli-CVG is not an established abbreviation, please use instead E. coli possessing VAGs or E. coli carrying VAGs or something similar

Lines 39-40: the part “such as the presence of eae, bfpB, and ial genes or stx2 and ial genes« is a bit confusing, please clarify is it the combination of genes? Or is it a mistake that ial genes were listed twice?

Line 45: Escherichia coli should be in italics, Bison bison should be in italics

Line 49-50: instead of microflora use microbiota and please add that E. coli lives with its host in mutualistic relationship.

Line 62: E. coli strains recovered from wildlife animals are unknown – please rephrase, as from some wildlife we have the data (e. g. brown bears)

Lines 65-67: please rephrase, as the sentences are not clear

Line 70: delete the word “essential”

Methods section -  please include the numbers of fecal samples, of analyzed E. coli in each step

Line 198: instead of intestinal flora use intestinal microbiota

Line 237, 241, 242: italicize the gene names

Line 244: italicize E. coli

Somewhere in the Discussion and in Conclusion it should be mentioned that further research is needed in order to establish whether the found E. coli possessing virulence-associated genes are actually possible to infect/be pathogenic for humans.

Comments on the Quality of English Language

Some minor English edits are needed through the paper.

Author Response

Reviewer 2

I appreciate your observations, and below I show you how they were addressed.

Throughout the paper – instead of virulence genes – use virulence-associated genes (VAG); E. coli-CVG is not an established abbreviation, please use instead E. coli possessing VAGs or E. coli carrying VAGs or something similar

The suggestion was accepted, and “virulence genes” were replaced by " virulence-associated genes.” Also, E. coli CVG was replaced by E. coli-CVAGs.

Lines 39-40: the part “such as the presence of eae, bfpB, and ial genes or stx2 and ial genes« is a bit confusing, please clarify is it the combination of genes? Or is it a mistake that ial genes were listed twice?

The text was modified to clarify that it is a combination (patterns) of virulence-associated genes in different strains.

Line 45: Escherichia coli should be in italics, Bison bison should be in italics

The words are already in italics.

Line 49-50: instead of microflora use microbiota and please add that E. coli lives with its host in mutualistic relationship.

The suggested corrections have been made. Specifically, "microflora" has been replaced with "microbiota," and it has been added that E. coli lives with its host in a mutualistic relationship.

Line 62: E. coli strains recovered from wildlife animals are unknown – please rephrase, as from some wildlife we have the data (e. g. brown bears)

The text was modified. The new paragraph states that the characteristics of E. coli strains isolated from free-range Bison are unknown. It also describes the imprecisions between phylogroup and virulence genes.

Lines 65-67: please rephrase, as the sentences are not clear

The unclear sentences in lines 65-67 have been rephrased for clarity.

Line 70: delete the word “essential”

The word “essential” was deleted.

Methods section - please include the numbers of fecal samples, of analyzed E. coli in each step

Number of samples was added in each step

Line 198: instead of intestinal flora use intestinal microbiota

The term "intestinal flora" in line 198 has been replaced with "intestinal microbiota”.

Line 237, 241, 242: italicize the gene names

Gene names were adjusted.

Line 244: italicize E. coli

E. coli in line 244 has been italicized.

Somewhere in the Discussion and in Conclusion it should be mentioned that further research is needed in order to establish whether the found E. coli possessing virulence-associated genes are actually possible to infect/be pathogenic for humans.

The suggestion was accepted, and the sentence “However, more research is needed to establish the zoonotic potential of E. coli-CVAGs” was added.